# Ecosystem Services and Linkages of Naturally Managed *Monotheca buxifolia* (Falc.) A. DC. Forests with Local Communities across Contiguous Mountainous Ranges in Pakistan

**DOI:** 10.3390/biology11101469

**Published:** 2022-10-07

**Authors:** Fayaz Ali, Nasrullah Khan, Oimahmad Rahmonov

**Affiliations:** 1Department of Botany, Shaheed Benazir Bhutto University, Sheringal, Dir Upper P.O. Box 18050, Pakistan; 2Department of Botany, University of Malakand, Dir Lower P.O. Box 18800, Pakistan; 3Faculty of Natural Sciences, Institute of Earth Sciences, University of Silesia in Katowice, 41-200 Sosnowiec, Poland

**Keywords:** socio-culture system, human well-being, cultural keystone species, *Monotheca* phytocoenosis, Hindukush, Suleiman Mountain ranges

## Abstract

**Simple Summary:**

*Monotheca buxifolia* is a wild fruit yielding tree species of the *Sapotaceae* family, frequently reported in northern and western parts of Pakistan. This species is highly involved in providing a number of services, including provisioning, regulation, maintenance and cultural indicating the close relations between society and the protection of mountain areas. Due to the poor socio-economic situation and natural hazards in this region, *Monotheca* phytocoenoses have been heavily exploited during the last few decades. Apart from the dominant species, we also report some other important tree species (*Juglans regia*, *Pinus roxburghii*, *Ficus palmata*, *Punica granatum*, *Olea ferruginea*, and *Acacia modesta*, etc.) that play a key role in improving the economic situation of the mountain inhabitants. These phytocoenoses should be properly managed on a priority basis to support long-term consumption.

**Abstract:**

The local community of the Suleiman and Hindukush mountain systems in Pakistan has largely depended on the natural resources of the environment since ancient times. The ecosystem of these regions is under huge pressure due to a lack of awareness and the uncontrolled interference of communal, commercial, security, political, and ecological conditions. The present study was designed to illuminate the link between mountain society and the consumption of the benefits from *Monotheca* phytocoenoses using the ecosystem services concept from the sphere of the socio-ecological system to cultural relations. The use of this approach is very important due to the visible role and dominant status of *Monotheca* vegetation within the ecological system of the region. *M. buxifolia* is strongly connected with both local and cultural traditions and is counted as a key species, particularly for high-mountain inhabitants. We report that *Monotheca* phytocoenoses provide several services including shelter, food, fodder, medicines, and wood, etc., to the indigenous community and is highly valued in the local culture because of the poor economic condition of the society. The concept of this cultural keystone species is crucial for understanding ecosystem services and must be considered for the protection and conservation of these habitats. The results of field and social studies have shown that the stable maintenance of *Monotheca* phytocoenosis forests ensures the existence of key species as the most important providers of ecosystem services, e.g., provisioning, regulation, maintenance and cultural services, indicating the close relations between society and the protection of mountain areas. According to the results obtained, the mountains community of the studied area believes that tree species like *M. buxifolia*, *F. palmata*, *O. ferruginea*, *P. granatum*, *A. modesta*, *J. regia*, etc., are the key components contributing to the function of both the mountain ecosystem and communities’ well-being. This approach will be extremely useful for ensuring an inclusive management of the socio-ecological system of the Hindukush and Suleiman Mountain ranges of Pakistan.

## 1. Introduction

Mountains host 12% of the global population and cover almost 24% of the total geographical land area [1]. Several biotic elements (flora and fauna) live in natural mountain habitats, often forming a refuge known as a hot spot of diversity [2]. A series of different elements including climatic, edaphic and anthropogenic factors collectively contribute to making the mountains a multilayered geosystem [3]. Mountains are generally characterised by an uneven relief, a unique variety of climatic conditions and water channels, conditioned by the altitude gradient and terrain exposure [4]. As a result, mountain ecosystems exhibit a great variety in the structure of services defined by the availability and rate of utilization by mountain inhabitants [5].

Mountain ecosystem products support the livelihood and well-being of 12% of the world’s population and play a prominent role in supporting human life, particularly in the lower reaches of high peaks [6]. The most common services include medicinal products, hay for grazing animals, habitats for flora and fauna, wild edible fruits, timber for construction and burning purposes, tourism, and pastoral spots, which are often the only source of life for mountain inhabitants [7,8]. Mountainous areas are frequently subjected to land use change (e.g., deforestation) to allow the cultivation of cereal crops in order to fulfil the needs of the ever-growing population [4,9]. Due to a number of factors including environmental condition, geographical location, and anthropogenic pressure, the services, structure, and function of mountain ecosystems are currently undergoing widespread changes [10].

A lack of basic health facilities, mass migration (due to the Afghan war), and population explosion has had and continues to have the most significant effect on the utilization of ecosystem services both in the northwest (Hindu Kush ranges) and in the southwest (Suleiman ranges) of Pakistan [11,12]. Moreover, the majority of Pakistan’s forests, especially those located in the north and west, belong to the local population and are managed according to local customs. These forests are critical sources of food, timber, wildlife, and identity for local communities, which makes their exploitation (both for woody and non-woody products) an easy task. The current climate scenarios along with the aforementioned factors are the key elements responsible for the poor condition (structure and function) of the mountain ecosystems [4]. Although forest management policies such as Joint Forest Management (JFM) exist in Pakistan to raise awareness among the local inhabitants about the sustainable consumption of ecosystem services in high mountain areas [13], they have failed due to inappropriately conventional tactics and a failure to engage the local community [14]. Other programs (the National Forest Policy of 2015 and the Billion Trees Tsunami Afforestation Project of 2018) were initiated in the northern region of the country [15] and later extended to the whole country because they significantly preclude the degradation of forests and have helped forest areas to remain comparatively stable for the last five years [16].

The global distribution of *Monotheca* ecosystems is very restricted. So far, they have only been reported in some areas of Pakistan, Afghanistan, Oman, and Saudi Arabia [4]. *Monotheca* phytocoenoses located in the Hindu Kush and Suleiman Mountain peaks of Pakistan are facing a huge pressure from natural and anthropogenic interference [17]. These habitats are highly involved in providing several ecosystem services (fruit, timber, shelter) to local inhabitants due to constant economic crises [18]. Plant species valued for their characteristics and services have long been considered a gift from the creator, one that should be managed with special care [19]. Both the canopy species and understory vegetation of *Monotheca* phytocoenosis provide fruits, food, fodder, traditional medicines, and spots for tourism [17]. *Monotheca*, along with their associates, are very important for mountain communities as they are symbols of culture and religious belief and are actively included in ceremonies, customs, and culinary traditions [20].

Keeping all this background in mind, the link of mountain inhabitants is visible with the socio-cultural and economic spheres of the region. Therefore, the current study was designed to present the linkages between the contemporary communities of the *Monotheca* mountains and the use of ecosystem services of high-mountain scrub vegetation. It also aimed to detect the potential factors (social, economic, environmental, and political) responsible for the changes in these services. The central part of our study was to consider the Hindukush and Suleiman Mountain ranges as a socio-ecological system, a cohesive system in which humans are a part of nature. The study used the concept of ecosystem services, which extends the scope of analysis of the socio-ecological system to the sphere of cultural relations of the contemporary society inhabiting the *Monotheca* mountains, traditional, indigenous knowledge, and cultural keystone species. It also sought to document the importance of keystone species for mountain inhabitants, which has been ignored in the valuation of ecosystem services.

## 2. Materials and Methods

### 2.1. Study Area

Pakistan is an important country in south Asia positioned from 60°55′ to 75°30′ longitude and 23°45′ to 36°50′ latitude covering an area of 803,943 km^2^ [17]. Diversity in climatic conditions and biota, with a reported 6000 vascular plant species, is due to a large altitudinal gradient ranging from sea level to Karakoram-2 (8611 m) [11]. Pakistan is categorized into five topographic zones, and the mountainous zone contributes almost 61% to the geographical area, accommodating nearly 40 million people out of 207.8 million [21]. Pakistan has four seasons, with reasonable variations in temperature both regionally and seasonally; regions toward the south are comparatively hotter and drier, while the northwest regions have a moderate climate [22]. In northern parts the temperature drops below freezing, reaching lows of −22 °C; however, the country touches 50 °C in southern regions during the summer. Similarly, a clear variation is observed in the precipitation rate, with 1500 mm to 2000 mm reported in the north and 100 mm to 200 mm in the south of Pakistan [23]. The local population of the studied area consists of Pushtoon, Baloch, and Chitralians mainly linked with farming, husbandry, gardening, cattle breeding, the lumber industry, and pastoralism. The poor condition of the soil cover, mainly due to steep slopes and poor irrigation systems, makes it difficult for locals to obtain crops and hay for cattle. In the studied area, the unemployment rate is comparatively high (65%); however, *Pinus*, *Acacia*, or *Eucalyptus* culture is a very common way of obtaining timber and other products as a source of income [24].

### 2.2. Scheme of Research

The methodology followed in this study documented the ecosystem services of *Monotheca* phytocoenosis in the high mountains area as a socio-ecological system [2]). It was mainly comprised of two subsystems i.e., socio-economic and ecological development. The basic set of methods for this study included a general survey, interviews, a spatial analysis, and analyses of source materials (Figure 1).

### 2.3. Recognition of the Diversity of Monotheca phytocoenoses

Field visits were arranged by an expert team in the summer seasons of 2018–2021, to document the ecosystem services within the phytocoenoses of *Monotheca* forests (Figure 2) located at an altitude ranging from 950 to 2472 m asl (above sea level) [4]. The northern and western slopes of the studied area are clearly different from each other in terms of insolation, moisture content, temperature, and edaphic properties [25], contributing to the formation of small but varied plant communities [26]. Both conifers (*Pinus* species, *Cedrus* etc.) and broadleaved tree species (*Acacia*, *Quercus* species, *O. ferruginea*, *F. palmata*, *M. buxifolia*, etc.) are the key components of these communities [27]. Similarly, understory species in the *Monotheca* phytocoenoses are mainly presented by *Dodonaea viscosa*, *Justicia adhatoda*, *Gymnosporia royleana*, *Saccharum species*, etc.

This study also focused on the exploration of floristic diversity with special reference to ecology, geography, and socio-economic impact. The consequences of the exploitation of natural resources (mainly keystone species) were also recorded through interviews with both shepherds and the mountain community. After a general survey and literature review, we assessed the floristic composition and diversity within the *Monotheca*-dominated ecosystems (Figure 3) according to commonly adopted methods [28]. Habitat conditions and absolute values such as density and crown cover were also calculated. Special attention was given to the occurrence of food plants present inside *Monotheca* ecosystems. The spatial distribution of *Monotheca* communities was analysed using the ArcGIS program based on satellite images from 2020 [2]. The studied regions are entirely positioned in the Hindukush and Suleiman Mountain ranges of Pakistan, showing four profiles from different directions due to altitudinal gradient and terrain exposition (Figure 4).

### 2.4. Identification of Cultural Key Species (CKS)

We identified the CKS via direct interviews with mountain inhabitants, forest officials, pastoralists, and other experts and reconfirmed them using relevant literature [29]. A questionnaire consisting of a set of different questions (Table 1) was designed for the identification of CKS following standard methodology [30]. Furthermore, we established a scale consisting of five levels (i.e., very high, high, moderate, low, and not used) to measure the importance of the traditional, ecological, and economic services of an individual indicator species. It is worth mentioning that the information about the consumption of plant products significantly varies within the society under investigation, depending upon the age and social group of respondents. In comparison to younger generations, older people are more familiar with authentic information about the uses of individual plant species [31]. Communities that are involved practically with plant species (pastorals, hakeems, gardeners, etc.) have a strong grip on traditional knowledge. Therefore, we decided that the potential of individuals will be measured qualitatively.

### 2.5. Identification of Ecosystem Services (ES)

We interviewed old people, traditional healers, local inhabitants, pastoralists, farmers, gardeners, and, more importantly, direct beneficiaries for the documentation of services from *Monotheca* phytocoenoses. We investigated the potential of services offered by an individual species through field visits to these ecosystems, available databases on several aspects of the environment (topography, soil and geological map, land use, demographic information, etc.), and information from the literature. The Common International Classification of Ecosystem Services (CICES version 5.1) with the amendment of Solon et al. [33] was used to document and analyse the ecosystem amenities of individual tree species including CKS. This system includes provisioning, regulation and maintenance, and cultural services. Overall, there were 8 divisions, 19 groups and 35 classes in this system).

### 2.6. Perception and Use of Ecosystem Services

Field visits and online surveys (via Microsoft’s platform) were performed to obtain information from mountain inhabitants on the perception and use of ecosystem services from *Monotheca* phytocoenoses [34]. An open-ended and semistructured questionnaire was designed, which anyone could fill in anonymously, with no need to register [35]. No limitation was imposed on the time taken to complete the questionnaire, which allowed respondents to think about their answers. Each respondent had the option to change the language of the form (in our case, the choice of English, Urdu, Pushto, or Kohistani was set automatically) [2]. The questionnaire contained three thematic sections: a—respondent’s information, b—traditional benefits, c—income opportunities, and d—medicinal information related to *Monotheca* phytocoenoses. Every respondent answered a total of 46 questions, including closed (25), open-ended (14), and multiple-choice questions (7). Respondents were selected randomly; however, preference was given to old people and those who were representatives of their households [31]. These criteria were followed to focus on provisioning services, which benefit a whole family, not a single individual.

## 3. Results

### 3.1. Diversity of Monotheca Forests

In the studied area, *Monotheca* phytocoenoses were clearly distinguished into four vegetation types (i.e., 1. *Mono-Olea*, 2. *Mono-Acacia*, 3. *Mono-Eucalyptus*, 4. Pure *Monotheca* crop). The average tree density in the first formation (Figure 5A) was 305.87 trees ha^−^^1^ with a basal area of 41.26 m^2^ ha^−1^. This group was characterised by altitudes (1328.1 m asl) and steep slopes (Table 2). Anthropogenic disturbances including overgrazing and land use change were the most visible disturbances reported in these *Monotheca*-dominated forests. The second formation, codominated by *Monotheca* and *Acacia* (Figure 5B,C), was recorded with a high tree density of 325.67 trees ha^−1^ and a basal area of 66.63 m^2^ ha^−^^1^. The forests of this group were located at low altitudes (1038.03 ± 68.8 m asl) and were characterised by less human pressure than group I.

Similarly, the ecosystems dominated by *Monotheca* and *Eucalyptus* formed the third vegetation type in the studied area and were located at a lesser altitude (1031 m asl) and moderate slope. Tree density for this group was 296.2 ha^−^^1^; however, due to the presence of large diameter trees, the basal area (95.93 m^2^ ha^−^^1^) was the maximum among all the reported vegetation types. The density of the *Monotheca* pure crop was 320 trees ha^−^^1^ with a basal area of 89.39 m^2^ ha^−^^1^ and signs of human interventions were reported. In these forests, the predominance of some plant species was due to environmental, topoclimatic, edaphic, and relief condition. Reported native (82%) and exotic (8%) tree species were mostly dicot (73%) with predominately megaphanerophytic lifeforms (88%), microphyll (33%), which largely reflect strong chronological differentiation, and distinct linkage (55%) to the Sino-Japanese phytogeographical region (Table 3).

The understory vegetation in the studied area was composed of a variety of herbs, shrubs and grasses, of which, *Dodonaea viscosa* was frequently observed (Figure 6A,B). *J. adhatoda*, *C. procera*, *Z. nummularia*, *P. rugosus*, *G. Royleana*, *V. negundo*, *N. ritchiana*, *L. inermis*, and species of *Withania* and *Solanum* were among the other shrubs reported in association with *D. viscosa*. Similarly, *C. oxyacantha*, *P. abrotanoides*, *M. parviflora*, *P. emodi*, *V. pilosa*, *U. dioica*, *C. murale*, *P. harmala*, *S. asper*, and others were common representatives from herbs. Grasses in the studied area consisted of *Sorghum halepense* and *Saccharum* species. The understories were found clear in some forests due to heavy anthropogenic interventions and site conditions in some areas (Figure 6C,D).

### 3.2. Keystone Species of Monotheca Forests as Ecosystem Services

We reported a number of edible plant species as an important component of *Monotheca* phytocoenoses contributing significant shares to the household of mountain inhabitants (Table 4). The occurrence of these plants (tree species *Olea*, *Monotheca*, *Acacia*, *Morus*, *Ficus*, *Punica*, *Ziziphus*, *Pinus* species, etc., along with shrubs and herbs) were observed in M. buxifolia forest zones often in diverse habitats which were often heavily exploited during winters. The consumption of these species for different purposes (spices, medicinal, food, fodder, income source, etc.) proves the close link to local tradition and inhabitants’ knowledge in this field. Plants such as *A. modesta*, and *O. ferruginea* are used in traditional ways to control blood pressure, while *Withania* and *Malva* species have been reported as a remedy for stomach diseases. According to the respondents, the fruit of Juglans is best as a brain tonic; therefore, the products of these CKS are harvested and consumed as a traditional medicine. This valuable knowledge is very important for high-mountains communities (deprived of basic health facilities) and is transferred to future generations verbally and through practices. In comparison to other species, the wood of *Monotheca*, *Acacia*, and *Olea* is durable and resistant to rot; therefore, it is used in households and as a fuel source. Apart from this, leaves are used as fodder while fruits are a good source of income in these rural areas. *A. lebeck*, *D. sissoo*, *E. globulus*, *G. optiva*, *M. azedarach*, *P. nigra*, *Q. baloot*, and *T. aphylla*, etc., present in these phytocoenoses are also important both for fuel and the making of agriculture tools.

Among the understories, *D. viscosa*, *J. adhatoda*, *G. Royleana*, *V. negundo*, *N. ritchiana*, etc., were the commonly occurring shrub species, and were actively involved in providing a variety of ecosystem services. Winter in these high mountain areas is very hard and the temperature often remains below the freezing point. To deal with this, the local community collects small leaves and twigs from high altitudes (in the summer season) then stored them and used them as fodder for their cattle. After the eating of leaves, the remaining twigs are more commonly used as fuel for bread ovens. Due to the recent population explosion and resource depletion, *Monotheca* mountains are heavily disturbed (Figure 7) for marble and coal extraction, making roads and fields for agriculture and horticulture cropping system (cash crops).

### 3.3. Ecosystem Services of Monotheca buxifolia

The Hindukush and Suleiman Mountain ranges are characterized along their entire length by a high diversity in climate, topography, and terrain exposure. This leads to the formation of a number of ecosystems (desert, grassland, scrub, forest) known for providing services (provisioning, regulation and maintenance, and cultural) to the mountain inhabitants. Monotheca forests provide habitats for a variety of flora and fauna, ensuring the maintenance of the life cycle, habitat, and genes, and are therefore rich in biodiversity. The forests of high altitudes are comparatively more protected due to their geographical position and have a greater capacity to provide services. The observance of the natural succession with no recent signs of soil erosion was the most promising sign linked to these high-altitude ecosystems. However, the forests at lower altitudes were comparatively more accessible and subject to more human influences, which have a negative effect on their richness, structure, and function.

We determined the provision of ecosystem services in different categories from Monotheca phytocoenoses using different sources (local experts, information from both direct and indirect beneficiaries, and research on vegetation), whose details are given in Table 5. Among the provisioning services, food, biomass, and energy were more prominent on the basis of consumption for the local community and were often counted as the only source of income for the household in the studied area. This section was used extensively particularly on the northern slopes due to the presence of green pastures and associated with the establishment of summer camps for grazing animals. Water is the most important requirement for livestock, so both seasonal and permanent shepherd’s camps tend to stay at the springs of water channels. Regulation and maintenance services are strongly linked with the natural renewal process of the environment and unfortunately, the mountain inhabitants have no idea that they are using these ecosystem services.

Social and economic changes showed a huge impact on the cultural ecosystem services in the entire region. In the past, local inhabitants visited these ecosystems for the collection of food, timber, medicinal plants, etc., but now, cultural services have been extended to include sport hunting, recreational fishing, hiking, climbing, traveling, mountain trekking, etc. This shift in the structure of cultural ecosystem services is related to a change in the economic situation of the locals. Presently, approximately 75% of the families from the studied areas (Districts = Sherani, Zhob, Parra Chinar, Musakhel, Bajaur, Dir, Malakand, Karak, etc.) have family members based outside the country to earn money. Although some time ago (20–30 years), the mountain communities collected fruit and timber on their own, currently they purchase them from the local market.

Native traditions, ceremonies, and religious features that have been transferred through generations constitute the spiritual and symbolic element of cultural services. Irrespective of social level, religious matters are accurately followed and respected by everyone in the studied area. The religious theme of nature conservation is perceived in the case of an individual tree. *O. ferruginea* plays this kind of role throughout the studied area, is frequently observed in Muslim graveyards, and is respected by everyone, while Monotheca is respected in some parts of the studied area (Parra Chinar, Zhob, Musakhel). The preservation of a specimen of this type is seen as God’s will, or as related to the existence of a Mazar, i.e., a sacred place. In the culture of the majority of Asians (and others), a Mazar is a place of repentance and reflection. People make pilgrimages to these places in order to shed their sins and to register their prayers and thanks.

### 3.4. Perception and Use by the Local Society of Ecosystem Services Provided by Monotheca Forests

The potential of actual local use (based on the respondents’ information) of ecosystem services from the *Monotheca* associations is presented in Table 5. A total of 572 individuals, categorized into two groups, based on direct and indirect beneficiaries (271 each), were interviewed (online surveys and direct discussion). The result of the chi-square statistics showed that the number of male respondents (76.53%) were recorded significantly different (χ^2^ = 0.0001) from female respondents (32.47%) (Table 6). This difference in gender number was attributed to a number of factors including religious, social, and cultural barriers. In total, 64.5% of the respondents represented medium household sizes having an average of 11 individuals followed by large (19%, >15 individuals) and small household size (11.64%). We reported a significant difference in the households of the respondents by performing the chi-square test (χ^2^ = 0.023).

We also assessed the socio-economic activities of the respondents from both groups (i.e., direct and indirect beneficiaries) and the results are displayed in Table 7. The largest group by occupation were the unemployed (21.96%), while others were engaged in different sectors (farming, pastoralist, drivers, petty business, government employees, etc.). Up to 12.18% of the respondents were students, followed by government employees (10.70%), arable farmers (9.96%), and livestock farmers (8.30%). The Chi-square statistics confirmed the significant difference between the respondents from different profession e.g., unemployed (χ^2^ = 0.0001), farmers (χ^2^ = 0.0006), day labourers (χ^2^ = 0.028), and livestock farmers (χ^2^ = 0.0018). The age of the respondents mattered greatly in this research as old people were able to provide accurate information due to practical experience. Table 8 presents the age of the respondents showing that young individuals were comparatively more involved in the collection and harvesting of fruit and timber from these phytocoenoses in comparison to old people. With respect to the literacy rate of the respondents, 43.73% lacked a basic education. The illiteracy rate was high and had the potential to affect the survey, but it is worth mentioning that the questionnaire was more accessible to illiterate people.

### 3.5. Provision of Services

According to 42.44% of the respondents, the most prominent fuel material was wood in the studied area followed by animal manure (31.92%) and hard coal (19.19%) (Figure 8). A total of 6.46% of respondents declared that they used other sources of fuel that were not mentioned in the survey. The collection of wood was mainly from orchards, while animal manure was collected from farms and pastures. *Monotheca* (52%) and *Acacia* (28%) species were largely used for fuel due to their long-lasting heat and high energy value. The use of wild fruiting trees such as *P. armeniaca*, *P. granatum*, *F. palmata* and *Monotheca* was very common among the locals. In hilly areas, 85% of the total firewood was harvested from high altitudes during the autumn to cope with the freezing temperatures during the harsh winters. The fuel was either collected by the respondents themselves (71%) or purchased (29%) from the local markets. As well as being used as fuel, the collected timber was also used for other purposes (fencing, construction, cultural practices).

The timber consumed by a household in a year could be transported by 20–30 donkeys, or 15–22 horses, while for some households (22%) these needs were fulfilled by more than 30 donkeys due to their large family size. The price of timber per donkey varied greatly among respondents depending upon the quality and quantity of wood and also on the distance of transportation. However, this price largely fell between PKR 1500 and 2200, i.e., around USD 117–160. Due to recent plantation projects (BTTAP), 7% of the respondents confirmed an increase in forest area. Up to 66% of the respondent claimed that the change in forest area had occurred in the last 10 or 15 years; however, some of the interviewees believed that these changes had become more visible during the last 5 or 10 years. Forests around the villages were found to be in comparatively good condition and according to 60% of the locals; this was mainly due to a reduction in the consumption of forest resources, fewer animals in pastures, migration to urban areas, and restrictions on illegal logging.

***Division:*** energy, group: biomass; ***class:*** plants and animals; ***Services:*** food (milk, fruit, and meat), species of *Acacia*, *Quercus*, *Morus*, *A. altissima*, *Monotheca*, *M. azedarach*, *C. decidua*, *D. sissoo*. The majority of respondents (86%) had domestic animals. Some of them (23%) bought hay for cattle, some (27%) obtained it from their land, and a portion (14%) used rented land to obtain food. The local community used information about the consumption of plants and their products both culturally and economically in order to provide food and earn money. Among trees, the commonly used food plants included: *Monotheca* (100%), *Juglans* (80%), *Morus* species (69%), *P. roxburghii* (67%), *P. armeniaca* (51%), *P. granatum* (43%), *Z. armatum* (19%), and *F. Palmata* (11%). Among shrubs and herbs, the most frequently used plants included: *D. viscosa* (74%), *C. macrophylla* (61%), *G. royleana* (49%), *I. gerardiana* (33%), *J. adhatoda* (31%), *R. dentatus* (27%), *W. somnifera, W. coagulans* (19%), *R. communis* (13%), *T. fruticans* (9%), etc. *G. optiva*, *A. nilotica*, *S. oleoides*, *Q. baloot*, *T. aphylla*, *B. papyrifera*, *A. lebeck*, etc., were also reported useful for locals. In total, 44% of the respondents consumed the documented plants as a source of medicines, followed by ingredient dishes (31%), spices (17%), and as drinks (8%).

The collection of wild edible fruits from these ecosystems was very common among locals. The current study reported that 42% of respondents collected fruits (Figure 9) for direct use while many of the locals harvested these fruits as an income source.

*Monotheca* was the dominant crop in these ecosystems and its medicinal uses are given in Table 9. Up to 35.6% of the locals utilized *Monotheca* products traditionally for the treatment of 11 different diseases. Similarly, these products were also reported to be beneficial for the treatment of digestive and urinary tract diseases and diabetes.

## 4. Discussion

### 4.1. Degradation of Monotheca Forests

The economy of Pakistan has experienced a significant growth over the last few decades due to free trade, resulting in a higher energy demand which has had some adverse effects on the natural environment [36]. Pakistan is an agricultural country, and its economy largely depends on this sector; however, due to the continuous population explosion, the booming need for agricultural land has led to massive deforestation [37]. The deforestation rate (2.1%) in Pakistan is the highest among the South Asian countries [38]. Apart from industrialization, some other factors, including global warming and floods (2010, 2011, 2022) have also heavily contributed to mountain degradation affecting forest habitats [39]. This degradation has pushed many economically important species (plants and animals) into the IUCN endangered category [40,41]. The degradation in *Monotheca* forests in the studied region was well documented by Ali et al. [38,41]. Similarly, in the current study, we reported a significant loss in the natural habitats of *Monotheca* forests due to marble and coal extraction (Buner District), land use change (Swat District), intensive felling (Musakhel District), poor management (Parra Chinar District), and overgrazing (Photo 4). Very poor natural regeneration is also a significant contributor to the loss of this economically and environmentally important tree species [17,41].

The importance of understory vegetation as an integral component of forest ecosystems cannot be denied. Understory vegetation is highly important in sustaining forest ecosystems’ composition, structure, and function, easing energy flow, maintaining the nutrient cycle, and supporting canopy development as a forest ecosystem driver [42]. Although the understory flora represents a comparatively small proportion of the total forest plant biomass, it explains a major part of the floral diversity [43]. We reported a number of different species of shrubs, herbs, and grasses which were actively involved in providing a variety of services (food, fodder, medicines, shelter) to local inhabitants. The protection of the forested area is the main concern of the forest department, which has established several programs to inform the locals about the importance of forests [44]. However, due to a lack of basic facilities, poor economic conditions, and unemployment, the mountain community continues to use these resources extensively [45]. People’s participation in the conservation and management of forests is of great importance, so the public must become involved in this global task.

### 4.2. Areal Specificity of Ecosystem Services

The use of natural resources and ecosystem services of *Monotheca*-dominated vegetation types are somewhat diverse from other mountain ecosystems of Pakistan [19,46]. This may be attributed to several factors including habitat characteristics, resources availability, characteristics of inhabitants, literacy rate, economic condition, mass migration, etc. Due to the rough terrain and steep slope, the studied *Monotheca* mountains located to the north (Hindukush) and west (Suleiman) were not accessible easily and were difficult for cultivation. However, areas located on the sides of water channels with proper irrigation systems had the potential for agricultural cropping. Similar to previous reports, the ecosystem services in the studied areas largely depended on wild-fruit-yielding tree species (*F. Palmata*, *P. granatum*, *J. regia*, *M. buxifolia* etc.), pastures (grasses), recreation sites (hunting, hiking), and spiritual and cultural sites [11,47,48,49]. The importance of wild edible-fruit-yielding tree species in high mountain areas have been described by several authors; however, a continuous decline in their population is an alarming situation for local inhabitants, because most often these are the only source of income and have the potential to uphold the life status of locals in sensitive ecoregions. The traditions, ritual, and beliefs of mountain communities also affect the protection of the ecosystem (Muslim graveyards) observed throughout the studied area [40,50]. Moreover, the purchase of timber (construction, heating material) is nowadays very difficult from local markets due to restriction policies and economic constrains. It really helps in the renewal of these biocenosis and related ecosystem services.

### 4.3. Current State and Tendencies Concerning to Relation between Socio-Ecological System and Ecosystem Services

The majority of the studied area has been facing a continuous war (Soviet-Afghan and US-Afghan war), which has led to an uncertain situation (economic, social, and political) and mass migration. It is important to note that from the start of the war until the present day, the inhabitants of these sensitive areas have been deprived of basic facilities (hospitals, education, electricity, employment, etc.). The unemployment rate makes the natural resources of the *Monotheca* forest ecosystem vulnerable, and these have been highly exploited by mountain inhabitants (both locals and migrants). Moreover, movement in these sensitive areas has been under observation, making it difficult to transport fuel material such as coal from neighbouring countries (Afghanistan, Iran), and thus these phytocoenoses are the only available source. Additionally, the wood quality (high energy, long-lasting heat, compactness, durability) of some tree species such as *Monotheca*, *Acacia*, *Olea* etc., from these ecosystems makes them more suitable for extra exploitation [51]. Some of the inhabitants collect wood from high altitude areas and bring it to the local market to be sold, which is the second stage of degradation of *Monotheca* vegetation. Additionally, wild edible fruiting trees and medicinal plants are used as a food source and are often the only source of income, which further contributes to their exploitation.

During the early 21st century, individuals from most families moved abroad to earn money, which changed the economic condition of the area and was something of a relief for the mountain ecosystem [52]. This move of young individuals to foreign countries left infirm and older people behind, who were unable to collect wood from high altitudes. It also aided in forest protection because wood collection is a very difficult task and requires physical strength and some special skills. Money from abroad has changed the economic condition of the studied area and most of the mountain societies have shifted from timber to other alternatives (CNG, LPG, and electricity) [53]. Reductions in livestock have also proved very helpful in the renewal of natural pastures [54] and natural regeneration in these phytocoenoses [37,43]. Some of the locals still dependent on livestock production practise seasonal migration through mountain areas, managing natural resources both for their own use and as an income source. However, this situation is now squeezed in comparison to earlier times when more individuals were involved in such activities to meet the needs of their families.

To fulfil the growing demands of the increasing population, land use change (deforestation) along with the development of irrigation channels was frequently observed in the studied area. As a result, there was illegal felling of forest trees. Currently, the collection of fruit and timber from *Monotheca* phytocoenoses to provide an income is common. Due to restrictions from forest departments, the wood traders set their own prices. Although, in comparison to the past, these activities are being carried out on a small scale, they are still a visible pressure on these ecosystems [55], particularly due to the low natural regeneration rate [17]. Apart from this, the depletion of the natural resources in the lower mountain reaches compels the locals to harvest plants at higher altitudes for wood, food, fodder, and medicinal purpose. Reaching higher altitudes requires more energy and skills, which leads to the consumption of other energy sources.

Due to the decline in the *Monotheca* forests, ecosystem services have been affected in the studied area. The degradation of mountain ecosystems leads to soil erosion, severe floods, and a loss of biodiversity, posing a serious threat to medicinal, edible, commercially, and ecologically important plant species [56,57]. It is easily concluded that both human and natural interference have a highly detrimental effect on ecosystem services. This statement is best supported by the rarity of some plant species (*Taxus wallichiana*, *Juglans regia*) which were very common in the past. The current survey reported male dominance in the use of ecosystem services due to the division of roles, which is very deeply linked to the social sphere. Social, religious, and cultural barriers prevent women from working and moving freely. However, it is very important to note that this had no effect on the current study because every respondent spoke about his own household, consisting of both women and men. Moreover, men are considered more appropriate respondents due to their practical involvement (access to mountains and local markets) in using ecosystem services [58,59,60]. The practical involvement of young people (<25 years) was recorded comparatively less than that of older individuals due to younger people’s comparative lack of knowledge.

### 4.4. Transformation of the Mountain Society and Ecosystem Services

The mountainous areas of Pakistan are under the influence of anthropogenic and natural interventions which adversely affect the ecosystem services of this sensitive ecoregion. The natural regeneration of *Monotheca* is very limited due to external (environmental) and internal conditions (seed). Once the *Monotheca* along with strong associates (*Acacia*, *Olea*) establishes itself as a young plant, it has the potential to survive for a long time (<1000 years). These tree species have the ability to sink more carbon in comparison to other woody tree species, which is due to the strong, compact, hard, and durable characteristics of their wood [4,11]. They also provide timber as a fuel and construction material, fodder, fruit, and traditional medicines, establishing a close link between high mountain ecosystems and communities [28,61]. Sometimes, these services are the only source of food for a household, so the degradation of these phytocoenoses is a time bomb for the loss of biodiversity and ecosystem services [19,62].

The recent decline (since the 1980s) in the number of pastures has significantly reduced livestock populations across the country [48]. Still, we observed some summer camps containing grazing animals, which definitely had a visible impact on the natural renewal of the *Monotheca* mountains [28]. The impact of grazing on the natural regeneration of ecosystems has also been reported by Fischer [63]. Due to developments in the tourism sector, the political and economic revolution, and the launching of different forest programs (e.g., BTTAP) in the studied area, some of the summer camps are equipped with basic facilities (e.g., water, roads) [64,65], but the majority of patches in these mountains remain poor in vegetation due to geographical features. The same situation is observed in the western part of the Pamir-Alay range [66,67].

Individuals as well as the government should take strong action to protect the structurally diverse *M. buxifolia* forests across their natural distribution range in Pakistan. Knowledge about these ecologically and commercially valuable forests needs to be extended so that people become aware of deforestation and its resultant consequences for wildlife, humans, and ecological systems. People’s participation in the conservation and management of forests is of great importance, so the public must become involved in this global task. Likewise, illegal logging should be banned, and afforestation programs such as the Billion Tree Tsunami Afforestation Project (BTTAP) must be launched on a larger scale, which might increase the population of these productive forests and would offer a great opportunity to deal with climate change impacts. The current results on the ecosystem services of *M. buxifolia* forests might be used to develop effective conservation strategies to achieve multifunctional specific goals related to biodiversity targets and sustainable forest landscapes in the long-term.

## 5. Conclusions

The ecosystem services of *Monotheca* mountains are severely affected by grazing, habitat degradation, and intensive felling both for construction and heating purposes.
*Monotheca*-dominated ecosystems have a visible impact on the socio-economic development of mountain societies by providing provisioning, cultural, maintenance, and regulatory services.These ecosystems form a mosaic of habitats and therefore, protect the biodiversity of the region.A change in the consumption of services in the Hindukush and Suleiman Mountain ranges of Pakistan was reported at the turn of the 20th and 21st centuries due to a number of factors such as the economic situation, the literacy ratio, and security risks (Pak-Afghan border).The reductions in shepherds’ camps and the comparatively high ratio of employment opportunities have had a positive visible effect on the natural renewal of these ecosystems.The strong link between the functioning of mountain communities and the use of the ecosystem services of *Monotheca* phytocoenoses forests in this area indicates the desirability of the partial replacement of the latter with alternative sources of services (provisioning: *Monotheca*, *Olea*, *Acacia*, *Eucalyptus* farming, use of hard coal) and the management of hard-to-reach areas (such as pasture, fodder).An increase in the use of cultural services (religious worship, traditions, and tourism) of these ecosystems as an income source has been reported, which may be grounds for optimism in the context of the protection of sensitive mountain ecosystems, but at the same time poses new challenges in terms of unsustainable tourism development.The concepts of ecosystem service and cultural keystone species are deeply rooted in a cultural and social context and provide a rationale for conservation management of the Hindukush and Suleiman Mountain ranges in Pakistan, combining the protection of biodiversity, the reinforcement of cultural identity, and the sustainable growth of society with a reliance on the assets of the natural surroundings.

## Figures and Tables

**Figure 1 biology-11-01469-f001:**
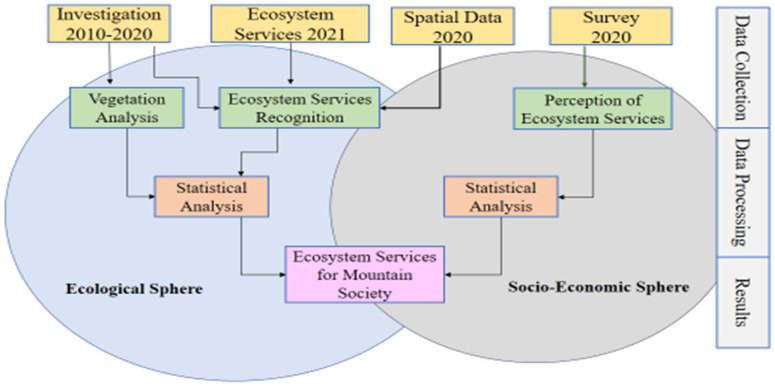
Methodology scheme used in assessment of *Monotheca* phytocoenoses forests ecosystem services.

**Figure 2 biology-11-01469-f002:**
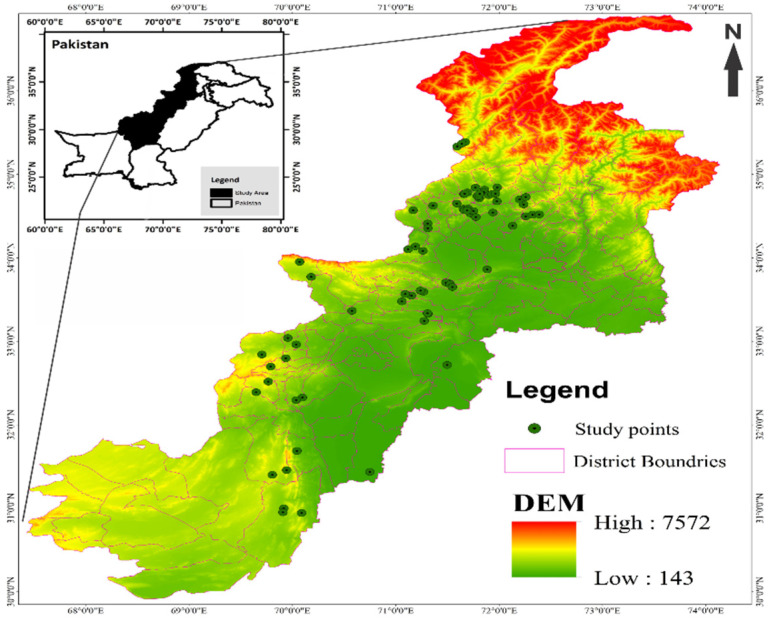
Distribution of *Monotheca*-dominated phytocoenoses in different elevation ranges across Pakistan.

**Figure 3 biology-11-01469-f003:**
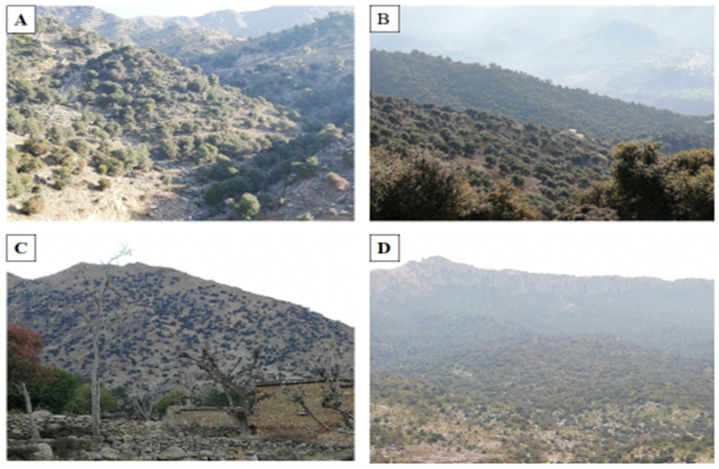
*Monotheca* forests at different locations in the study area, (**A**). Dir District, (**B**). Malakand District, (**C**). Bajaur District, (**D**). Musakhel District.

**Figure 4 biology-11-01469-f004:**
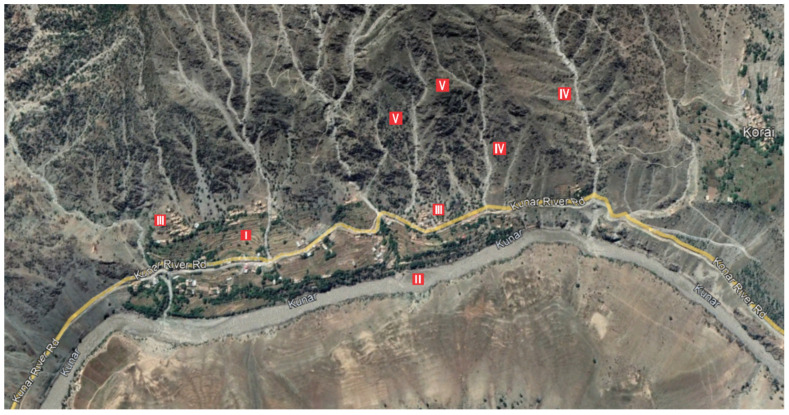
*M. buxifolia* forest sampling location at Pak-Afghan border envisage (Hindukush range) I. agriculture land, II. Kunar River, III. local community, IV. meander streams, V. *Monotheca* forests.

**Figure 5 biology-11-01469-f005:**
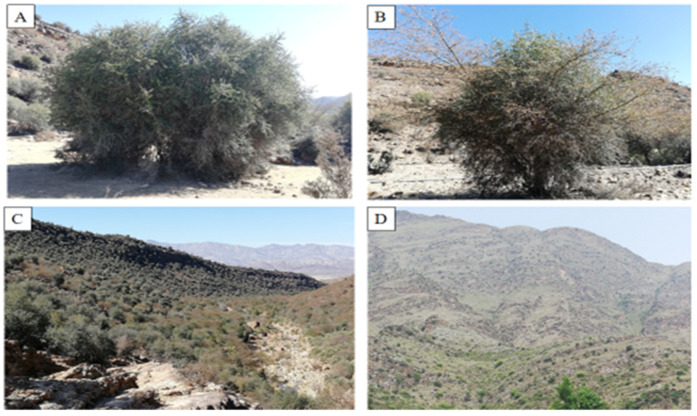
(**A**) *Monotheca*-*Olea* phytocoenosis; (**B**) *Monotheca*-*Acacia* phytocoenosis; (**C**) *Monotheca*-*Olea*-*Acacia* mixed forest; (**D**) very scarce vegetation of *Monotheca*.

**Figure 6 biology-11-01469-f006:**
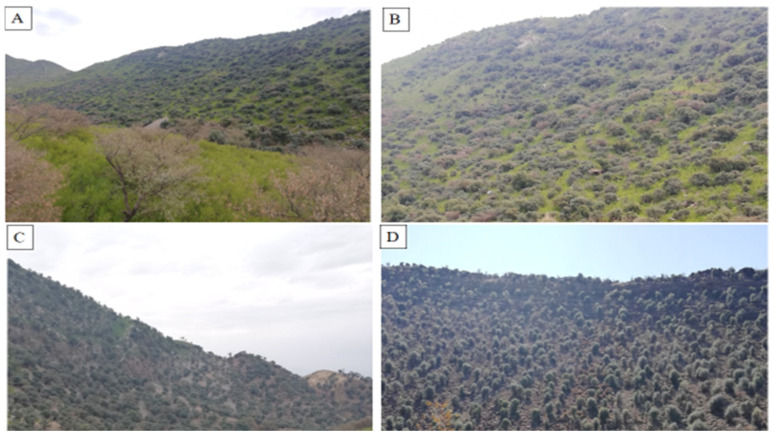
Understory vegetation of *Monotheca* forests. (**A**,**B**) Understory dominated by *Dodonaea viscosa*, (**C**,**D**) clear understory vegetation.

**Figure 7 biology-11-01469-f007:**
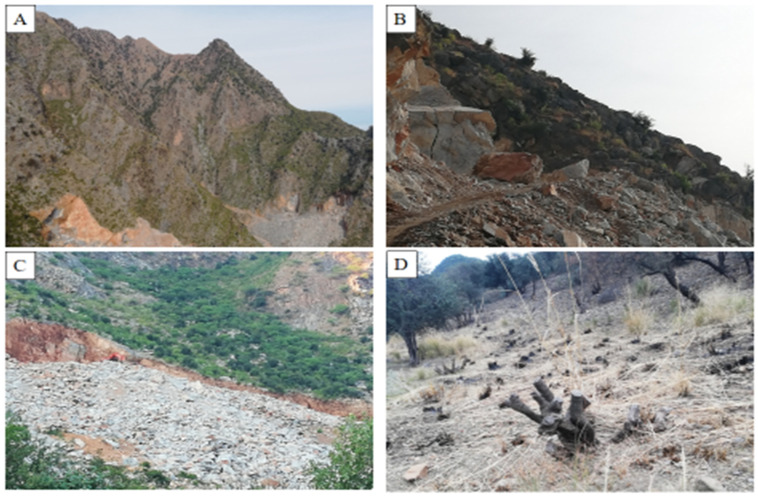
Disturbances in *Monotheca* forests: (**A**) marble extraction in district Buner, (**B**) making of roads in Khyber District, (**C**) land use change in Sherani District, (**D**) Cutting in Nowshehra District.

**Figure 8 biology-11-01469-f008:**
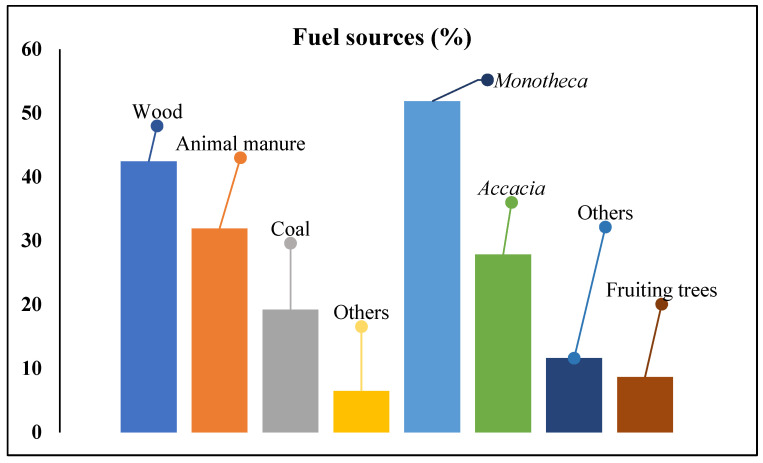
Different sources (column 1 to 4) used as a fuel material in households. Types of trees (column 5–8) consumed as fuel material.

**Figure 9 biology-11-01469-f009:**
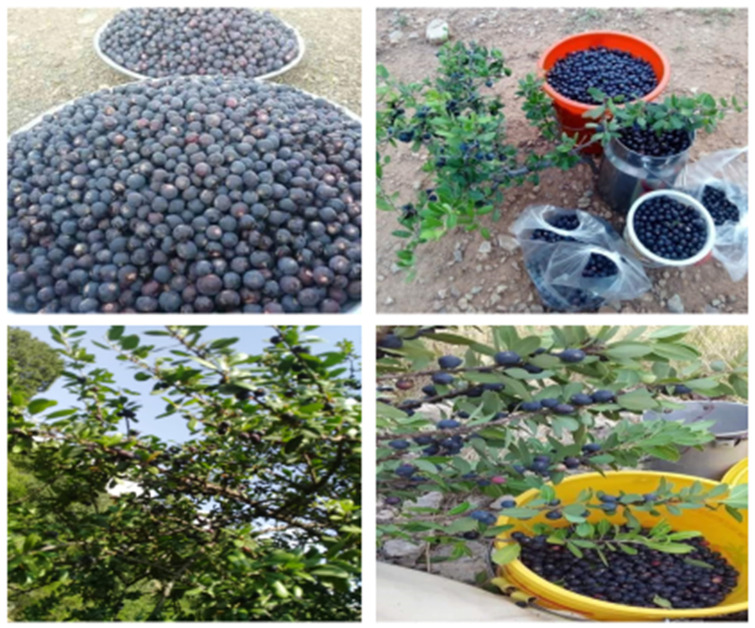
*Monotheca* fruits in the local market.

**Table 1 biology-11-01469-t001:** Questions for reporting cultural keystone species following Rahmonov et al. [2] and Garibaldi and Turner [32].

S.no	Questions
1	What is the rate and procedure of consumption of species? Can the species be used for more than one purpose?
2	Is there any specialized vocabulary or names for the species in the native language?
3	Is the species involved directly or indirectly in ceremonials, songs, and symbols?
4	Is the species still alive in the minds of locals irrespective of social, cultural, and economic changes?
5	What are the chances for the replacement of CKS with other species (native)?
6	Is there any economic benefit from these species?

**Table 2 biology-11-01469-t002:** Phytocoenotical differentiation of *M. buxifolia* associations based on densitometric and topographic variables.

Forest Type	Dominant Species	Altitude (m asl)	Density (ha^−1^)	Crown Cover (m^2^ ha^−1^)	Slope	Aspect Degree
**I**	*M. buxifolia*, *O. ferruginea*, *A. modesta*, *M. alba*, *F. palmate*, *Z. mauritiana*	1328.1	305.87	41.26	28.5	203.3
**II**	*M. buxifolia*, *A. modesta*, *A. altissima*, *P. roxburghii*, *P. granatum*, *C. australis*	1038.03	325.67	66.63	21.1	225.75
**III**	*M. buxifolia*, *E. globulus*, *A. nilotica*, *P. nigra*, *C. decidua*, *T. aphylla*	1031.72	296.2	95.93	23.25	219.3
**IV**	*M. buxifolia*, *Z. mauritiana*, *P. dactylifera*, *P. granatum*, *M. alba*	1275.3	324.7	44.97	26.79	254.93

**Table 3 biology-11-01469-t003:** Summary of chorologies, tree type, life form, and leaf spectra of the documented tree species.

**Chorologies**							
**Chorotypes**	**CCS (%)**	**Chorotypes**	**CCS (%)**	**Chorotypes**	**CCS (%)**	**Chorotypes**	**CCS (%)**
SU	11.5	EA	5.77	GR	1.92	M	5.77
SS	5.77	ES	15.38	SA	5.77	Cos	1.92
IT	23.08	Trop	7.69	PLU	5.77	CA	3.85
Cir	1.92	ITL	1.92	IGB	1.92		
Tree Types, Life form and Leaf spectra							
**Tree type**	**CCS (%)**		**Life Form**	**CCS (%)**	**Leaf Spectra**		**CCS (%)**
BLD	59.26		MP	88.89	L		22.22
BLE	29.63		NP	11.11	Mi		33.33
CE	11.11		---	---	Me		25.93
AD	88.89		---	---	AP		3.70
AM	3.70		---	---	Na		14.81
GM	7.41		---	---	---		---

Explanations: CCS, contribution in current study; SU, Sudno-Zambezian; SS, Saharo-Sindian; IT, Irano-Turanian; CA, Central Asian, EA, East Asian; ITL, Italian; SA, Saharo-Arabian; PLU, pluriregional; IGB, Indo-Gangetic Basin; ES, Euro-Siberian; M, Mediterranean; Trop, tropical; Cos, cosmopolitan; BLD, broad-leaved deciduous; BLE, broad-leaved evergreen; AD, angiospermic dicot; AM, angiospermic monocot; CE, conifer evergreen; GM, gymnosperm; MP, megaphanerophyte; NP, nanophanerophyte; L, leptophyllous; Mi, microphyllous; Me, mesophyllous; Ap, apophyllous; Na, nanophyllous.

**Table 4 biology-11-01469-t004:** Cultural keystone species and their potential for providing ecosystem services to the mountain communities.

*A. modesta* (trunk, leaves, roots)	Very high	Protect soil washing, soil-forming factor, fodder for cattle	Fuel, construction material	Agricultural tools (axe, wedges etc.), attract honeybees	Common	High
*Ficus palmitate* (fruit, roots)	High	Inhibition of soil erosion, human and wildlife food, soil formation	Fuel	Traditional meals	Common	High
*Juglans regia* (trunk, fruit, roots)	moderate	Provisioning, regulation, culture, habitat for wildlife	Timber (firewood, building material), fruit is edible	Medicinal plant (brain tonic), burials (protection of body)	Low (in specific areas)	Low
*Monotheca buxifolia* (wood, leaves, fruit, roots)	High	Provisioning, regulation, cultural: inhabitation of soil erosion, wildlife food, cover, nesting habitat, maintaining biodiversity, soil formation, leaves as forage by grazing animals	Timber (firewood), fencing around cultivated fields	Medicinal plant (blood purifier, analgesic, digestive, laxative, healing, urinary track disorders), compensates for iron deficiency	High	High (exploitation of this species as an income source and as fodder put it in danger position
*Olea ferruginea* (stem, branches, leaves, fruit, roots)	Very high	Provisioning (fruit, fauna), fuel, regulation and maintenance (paedogenesis, home for fauna and birds)	Fuel, construction material	Furniture and agricultural tools, medicinal plant (antidiabetic, antiseptic, tonic)	Common	High
*Prunus armeniaca* (Timber, fruit)	High	Provisioning (fruit) and maintenance	Fuel	Fruit if helpful in digestion and also found helpful for blood purification	Common	Low
Shrubs and herbs						
*D. viscosa* (branches, leaves)	Very high	Regulation, soil binder, help in succession, serves as fodder; the dried plant is used as a fuel	Branches are used in thatching and hedging	Ethno-veterinary (leaves are astringent for goat rheumatism)	High	High
*Justicia adhatoda* (branches, leaves)	High	Regulation, soil binder, help in succession	Fuel	Leaves are used as an expectorant and anti-inflammatory agent.	High	Moderate
*Nannorrhops ritchiana* (leaves, branches)	Low	Cultural and regulation ecosystem services	Thatching material	The decoction of leaves is used for stomach problems	Rare	Very low
*Malva parviflora* (whole plant)	Low	Soil binder, dried plants are used for different medicinal purposes	Forage for grazing animals	Laxative and good in stomach problems	Rare	Low
*Withania coagulans*, *W. somnifera* (leaves, fruit)	Rare	Provisioning and regulatory services: prevent soil erosion, provide a home to newly emerged species	Leaves and fruits are available as fodder for grazing animals	Powdered leaves are good for stomach pain. Fruits are given for urinary track inflammation	Low	Rare

**Table 5 biology-11-01469-t005:** * Ecosystem services provided by *Monotheca* forests in Pakistan.

Section	Division	Group	Class	Species/Service
Provisioning	Nourishment	Biomass	Crops	Edible-fruit-wielding tree species such as *Monotheca*, *Punica*, *Olea*, *Juglans*, *Celtus*, *Z. mauritiana*, *Ficua* and *Morus*
			Minerals	Asphaltum (rich source of minerals)
			Farm animals	Livestock products (Meat, milk), honey
			Wild plants	*P. granatum*, *A. nilotica*, *Z. armatum*, *G. royleana*, *I. gerardiana*
			Wild animals	*Capra falconeri*, *Ovis* *Vigneii*, *Canis aureus*, *Panthera pardus*, *Ursus thibetanus*
			Wild birds	*Falco peregrinus*, *Alectoris chukar*, *Ammoperdix griseogularis*
		Water	From surface source	River Panjkorra, Chitral, Hingol, Dab, Kingri and their attributes and glaciers
			From underground sources	Water sources below ground in *Monotheca* habitats
	Constituents	Biomass	Fibres and materials for agriculture	Wood (construction, fuel), medicinal plants (*B. lyceum*), wool (sheep), fertilizers (animal manure as fuel)
	Energy	Biomass-based energy	From plants	Firewood and energy crops (*Acacia* and *Ziziphus* species, *M. azedarach*, etc.)
			From animals	Animal excretory products
		Mechanical energy	Animal work	Use in different agriculture practices
Regulation and maintenance	Mediation of waste, toxins, and other nuisances	Mediation by biota	By microorganisms, plants, and animals	Decaying and mineralization in the soil, purification of air by forest trees
			By mechanical process	Protection of soil cover from erosion
		Arbitration	By mechanical process	Organic material accumulation in ecosystems
	Mediation of flows	Mass flows	Slowing of mass movement	Stabilization of oil cover (roots, soil binder)
		Liquid flows	Water regulation	Regulates water flow across the slopes, flood protection
		Gaseous flow	Protection from air storms	Mountain peaks and valleys
			Ventilation and transpiration	Mountain peaks and valleys
	Maintenance of physical, biological, and chemical conditions	Life cycle maintenance, habitat, and gene pool protection	Pollination and seed dispersal	Agents for pollination and dispersal of seeds
			Habitat maintenance	Provides habitat, biogeographic regions for speciation, and hot spots for biodiversity
		Soil formation and composition	Airing	Maintain soil physicochemical properties
			Matter decomposition and assimilation of elements	Maintenance of soil biochemical properties through natural factors
		Water status	Chemical properties	Composition of parent rocks
		Climatic conditions	Microclimate regulation	Uneven terrain from moderate to steep slopes, U- and V-shaped alleys
Cultural	Interactions (physical)	Physical and experimental interaction	Sports and refreshment	Tracking, hiking, climbing, riding, and hunting.
			Natural observation (watching)	Watching birds, rare mammals, and unique natural elements
		Intellectual and representative interactions	Research, academics, and creative work	Exploration of flora and fauna, climate change, environmental monitoring
	Spiritual, symbolic, and other interactions with biota, ecosystems, and lands/ landscapes	Spiritual or emblematic	Shaping identity	Plants for cultural practices (*O. ferruginea*), emblematic plants (*P. gerardiana*), animals (*Capra falconeri*, *Ursus thibetanus*), and birds (*Falco peregrinus*, *Alectoris chukar*).
			Shaping attitudes	Holy places; sacred plants (*O. ferruginea*) and animals and their parts
		Other cultural outputs	Spiritual	Visiting places of spiritual worship (old graveyards in Pakistan), sacred trees (*M. buxifolia* near water resources and away from community)
			Existential	Longing for undisturbed forests as sources of ecosystem services

Note: * reported services were classified following Common International Classification of Ecosystem Services (CICES version 5.1) with the amendment of Solon et al. (2017).

**Table 6 biology-11-01469-t006:** Gender distribution and household size of the respondents.

Variables	Community Status		Total	χ^2^
	Directly Involved (*n* = 271)	Indirectly Involved (*n* = 271)		
Gender				
Male	229 (84.50)	183 (67.53)	412 (76.53)	0.00001 *
Female	42 (15.50)	88 (32.47)	130 (32.47)	
Household Size				
1–7	22 (8.12)	41 (15.13)	63 (11.62)	0.2236 *
8–14	184 (67.90)	166 (61.25)	350 (64.58)	
15–21	48 (17.71)	55 (20.30)	103 (19.0)	
>21	17 (6.27)	9 (3.32)	26 (4.80)	

* Significant at *p* < 0.05.

**Table 7 biology-11-01469-t007:** Socio-economic activities of the respondents. Figures in parentheses are averages while those outside of parentheses are frequencies.

Variables	Community Status		Total	χ^2^
	Directly Involved (*n* = 271)	Indirectly Involved (*n* = 271)		
Main occupation				
Unemployed	33 (12.18)	86 (31.73)	119 (21.96)	0.00001 *
Farmers	37 (13.65)	17 (6.27)	54 (9.96)	0.0064 *
Livestock	33 (12.18)	12 (4.43)	45 (8.30)	0.0018 *
Drivers	11 (4.06)	15 (5.54)	26 (4.80)	0.5464 ^ns^
Government employees	27 (9.96)	31 (11.46)	58 (10.70)	0.6767 ^ns^
Foresters and gardeners	7 (2.58)	4 (1.48)	11 (2.03)	0.5423 ^ns^
Petty business	19 (7.01)	18 (6.64)	37 (6.83)	0.8647 ^ns^
Labourers	26 (9.23)	12 (4.43)	37 (6.83)	0.0287 *
Housewives	13 (4.80)	30 (11.30)	43 (7.93)	0.0110 *
Students	40 (14.76)	26 (9.59)	66 (12.18)	0.0877 ^ns^
Others	26 (9.59)	20 (7.38)	46 (8.49)	0.4409 ^ns^

* Significant at *p* < 0.05, ^ns^ = nonsignificant.

**Table 8 biology-11-01469-t008:** Educational status age categories of the respondents.

Variables	Community Status		Total	χ^2^
	Directly Involved (*n* = 271)	Indirectly Involved (*n* = 271)		
Educational status				
Illiterate	109 (40.22)	128 (47.23)	237 (43.73)	0.00017 *
10 years	73 (26.94)	36 (13.28)	109 (20.11)	
10 to 12 years	41 (15.13)	34 (12.55)	75 (13.84)	
>12 years	48 (17.71)	73 (26.94)	121 (22.32)	
Age categories				
<20	48 (17.72)	17 (6.28)	65 (11.99)	0.00060 *
20–30	55 (20.30)	75 (27.68)	130 (23.99)	
30–40	62 (20.88)	66 (24.35)	128 (23.62)	
40–50	54 (19.93)	38 (14.02)	92 (16.97)	
>50	66 (24.35)	61 (22.51)	127 (23.43)	

Figures in parentheses are averages while those outside of parentheses are frequencies. * Significant at *p* < 0.05.

**Table 9 biology-11-01469-t009:** Medicinal uses of *Monotheca* products by local community.

Diseases	Number of Respondents	Way of Usage	%
Analgesic	21	Fruit is directly taken. In some area (Baluchistan), Fruit is boiled in water then mixed with oil (Desi ghee) and taken with bread.	10.88
Blood production	18	Fruit is directly taken.	9.33
Constipation	11	Fruit is directly taken.	5.70
Diabetes	22	Leaves and young branches are boiled in water, then use the water for drinking.	11.40
Digestive	33	Fruit is directly taken.	17.10
Dysentery	8	Fruit is directly taken.	4.15
HCV	9	Leaves and young branches are boiled in water, then use the water for drinking.	4.66
Healing	20	Powder of leaves and wood are kept on external wounds for healing.	10.36
Laxative	19	Fruit is directly taken. Leaves and young branches are boiled in water, then use the water for drinking.	9.84
Urinary track diseases	23	Leaves and young branches are boiled in water, then use the water for drinking.	11.92
Worms	9	Leaves and young branches are boiled in water, then use the water for drinking.	4.66
Total	193		100

## Data Availability

Not applicable.

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
