# Peer review of "Ecosystem Services and Linkages of Naturally Managed Monotheca buxifolia (Falc.) A. DC. Forests with Local Communities across Contiguous Mountainous Ranges in Pakistan"

_biology, 2022, doi:10.3390/biology11101469_

Round 1

Reviewer 1 Report

Dear authors, 

Thank you very much for the opportunity to read a very interesting and relevant article on ecosystem services associated with ‘traditional’ mountain forests in Pakistan. 

A general comment regards the need for a revision of the English language. Please check the notes below. They also contain some minor notes regarding some doubts which emerged while reading the text, which need to be clarified or corrected. 

Line 22 - ‘depends’ instead of ‘de-pends’

Line 24 - ‘de-signed’

Line 25 - Please check if the word ‘ingesting’ is the right word for this phrase. 

Line 31 - ‘com-munity’

Line 37 - ‘… maintenance and cultural

Line 52 - ‘con-tribute’

Lines 62-65 - Agriculture shows anywhere where there are people, and mountains are no exception.  Please review and clarify your point in the phrase “Mountains areas are not suitable for growing agriculture and cereal crops due to rough terrain, uneven slope, and poor irrigation system,…”. 

Lines 77-78 - “Although forest management policies exist in Pakistan but, they failed due to conventional tactics and ignoring the local community”

Lines 77-81 - The two phrases seem somehow contradictory. First, you mention that “forest management policies failed due to (…) ignoring the local community”, but then you mention that the JFM forum was established “to bring awareness among the local inhabitants”. Please clarify. 

Line 91 - “in” repeated

Line 95 and other references along the text - “Monotheca phytocoenosis” should be in italic

Please revise all species’ names in Latin to ensure they are in italic.  

Line 100 - word missing: “Therefore, we design the current study in” 

Line 129 - Figure 1 shows sampling locations of M. buxifolia, but there is no mention of this in the text before the figure. Maybe add a reference to Figure 1 after mentioning the altitude range in the paragraph in lines 148-150. 

Line 164 - “assessed” instead of “accessed”. Similar in the first line after Table 6 (no line numbers). 

Line 206 - Not clear, do you mean “…were used to analyse the documented ecosystem amenities…”, or “…were used to analyse the and document ecosystem amenities…”?

Lines 212-223 - Please revise the wording (e.g. “allowed them to about their answers “). 

Line 292 - “(deprived from basic health facilities)”

NO LINE NUMBERS AFTER Table 4

FIRST paragraph in section 3.3.

Please review the phrase for a clearer meaning: “Due to steep slope and no accessibility, the forests of high altitudes were highly protected and having a high potential of ecosystem services.”

THIRD paragraph of section 3.3.

Regarding the phrase: “But now, the cultural services have been extended to hunting, fishing, hiking, climbing, travelling, mountain trekking etc.” 

I imagine hunting and fishing were always present in local communities as basic ‘survival’ activities to supply food to the families, so maybe you intend to refer to ‘recreational or sport hunting’ and ‘recreational or sport fishing’? Please clarify. 

Regarding text in section 3.5., for clear reading, I would suggest adding the first phrase corresponding to the ES division to be formatted as a subtitle, e.g., bold + italic. Otherwise, it is confusing. Apply to the subsequent paragraphs addressing a new division. 

E.g. using…

Division: nourishment, group: energy sources (biomass-based); class: plant and animals as energy sources; Services: fuel material (wood, manure). 

According to 42.44 % of the respondents, the most prominent fuel material is wood in the studied area followed by animal manure (31.92%) and hard coal (19.19%) (Figure 4). (…)”

Instead of…

“Division: nourishment, group: energy sources (biomass-based); class: plant and animals as energy sources; Services: fuel material (wood, manure). According to 42.44 % of the respondents, the most prominent fuel material is wood in the studied area followed by animal manure (31.92%) and hard coal (19.19%) (Figure 4). (…)” 

Paragraph after Figure 4 - Confusing as it mentions wood requirement as fulfilled by donkeys.  (“The annual requirement of wood is full filled fulfilled by 20-30 donkeys, or 15-22 horses, while for some households (22%) this these needs were full filled fulfilled by > 30 donkeys”). Are you referring to the use of animal manure as fuel source? Please clarify. 

Please also revise Figure 5 regarding this issue, and provide more clear titles for each plot. Please clarify what you mean by ‘Price of timber per donkey’. 

Figure 6 - The plot is not clear. Are they percentages? Please revise the plot. 

In Section 4.2. - “…the eco-system services in the studied areas largely depend on wild fruit-yielding tree species…”

In section 4.3. - “… the inhabitants of these sensitive areas are still avoided missing basic life facilities…”

Please revise the English for this particular section. 

Last phrase of section 4.4. “Future research directions may also be highlighted” should be turned into a short proposal for research directions. 

Author Response

Thank you very much for your REVIEW on our manuscript entitled “Hidden ecosystem services and linkages of the naturally man-aged Monotheca buxifolia (Falc.) A. DC. forests with local communities across contiguous mountainous ranges in Pakistan ". The paper has been duly revised according to the comments made by you. We would like to submit the revised version for consideration of possible publication as a regular paper. We would like to express our sincere appreciation to you for your constructive comments and the effort and the time spent helping us to improve the quality of the paper. In the following, we provide a specific response to the comment, explaining how the paper is revised. We have highlighted all the changes in our revised manuscript by red color. We hope that the revised manuscript addressed your concerns in a proper and satisfactory way.

Review in an attachment

Reviewer 2 Report

The authors present an interesting study about an interesting region of Pakistan were there are no many information about some forest’s ecosystems. Combining several methodologies (field work and questionnaires) the authors deals with the ES provisioning in Monotheca type and identify the current situation and potential threats.

The added value of the study is the ecosystems analyzed and the difficult to work with certain human communities.

I think that the manuscript could arise the interest of a broad audience.

The research is well designed, the methodology is appropriate and the results and conclusions fit the objectives of the study. The extent and quality of the bibliography is adequate.

Anyway, there are some issues that should be improved in order to become a paper for Biology journal. To the margin of the comments below, the manuscript should be restructured in order to follow a logic and clear thread.

My sense is that the second part of the manuscript, since discussion block, is better organized than the first (introduction, methodology and results)

GENERAL ASPECTS

Title is not clear; the concept of hidden ecosystems is a little bit strange.

The authors use Monotheca concept many times without italics. Is necessary to correct this. Check the whole manuscript to be sure that all the names of the species are in italics.

The concept of micro-ecosystem referring to the Monotheca forests is a little bit strange for me. I’ll try to avoid this.

Maybe one of the options is to use the acronym MB (or something like that) to be   

INTRODUCTION

In line 59 you indicate the ecosystem services provided by the mountains. Here you should add some sentence about that.

Line 69, change “very common” by another word.

In line 74 you indicate that forests belong to the local population, this should be explained conscientiously because it seems interesting for the audience.

Line 88 needs a cite (Monotheca distribution).

Line 91 there are two “in”.

The last sentences of introductions are a little bit confused, the goal of the study (change in the ecosystem service of the consumption of high mountain scrub vegetation?) is not clear.

MATERIAL AND METHODS

Figure 1 should improve. A Regional map of Pakistan in order to locate the study area is needed. The caption should indicate that is a topographical map.

Line 133, the name of the genus should be in italics.

In line 145 you mention that the studied specie is Monotheca buxifolia. It should be mentioned in introduction; there you indicate Monotheca phiytocoenosis !!

In line 179 you indicate the process of CKS indication but in the questionnaire the questions are so basic then I’m not sure if you should have identified a list of species previously in order to avoid a great variety of these.

Line 211. This block is not well explained, is necessary be more concise on the methodology of selection of the inhabitants.

RESULTS

In table 2 (line 241) you indicate a set of features of the four vegetation types that are not explained in methodology block.

Table 4 is very interesting cause explains the ecosystem services of the keystone species of the studied area. The name of columns indicating what is what is necessary.

The block 3.4 includes a statistical analysis (chi-square), for me is not necessary here because to explain gender and household size differences with the % is sufficient. Anyway, if the authors prefer to keep this analysis, a block named “statistical analysis” is necessary in methodology block.

Figure 4 is interesting but by itself is not well understood. Should improve in order that be more understandable. Or maybe is not necessary.

Figure 5 is not needed due to the information is in the text.

Text in the page 18 and figure 6 should be carefully revised. Consider do not add the figure 6 due to is not clear. Also consider take out figure 7 because is not clear either.

DISCUSSION

Take out the reference of WWF (2018) and Ali et al. from the text.

Take out the last two sentences of this block. They are the recommendations of the contents of the block!!

CONCLUSIONS

In point 5 put the name of the tree’s genus in italics.

That’s all.

I’ve enjoyed reading your manuscript, congratulations for the research carried out.

Author Response

(The authors gave the same response as above.)

Round 2

Reviewer 2 Report

Dear authors, 

You have done a good job and have incorporated most of the provided suggestions.

Anyway, there are a couple of aspects that should be revised. They are easy.

1. Map (figure 2) should improve. The caption is not clear and I'd add a map of Pakistan in order to locate the study area. Remember to indicate that is topographical map.

2. The Figure 4 has improved notably but is necessary to sort the columns from largest (Monotheca) to smallest (other).

For me, with this couple of minor changes the manuscript is ready to become a paper for Biology journal.

Congratulations again.

Author Response

Response to Reviewer 2 Comments

Comments and Suggestions for Authors

The authors present an interesting study about an interesting region of Pakistan were there are no many information about some forest’s ecosystems. Combining several methodologies (field work and questionnaires) the authors deals with the ES provisioning in Monotheca type and identify the current situation and potential threats.

Point 1: You have done a good job and have incorporated most of the provided suggestions.

Response 1: We are thankful to the anonymous reviewer for keenly studying our manuscript and encouraging us for scientific research in this field.

Point 2: Map (figure 2) should improve. The caption is not clear and I'd add a map of Pakistan in order to locate the study area. Remember to indicate that is topographical map.

Response 2: The map and captioned is revised as suggested by the reviewer.

Point 3: The Figure 4 has improved notably but is necessary to sort the columns from largest (Monotheca) to smallest (other).

Response 3: Respected reviewer, this figure contains two different information as mentioned in the captioned. 1st portion (column 1 to 4) describes general source of fuel in the area. While the 2nd portion (column 5 to 8) focused on the forest trees as fuel source. Therefore arrangement from largest to small will be wrong.